# Urinary Biomarkers in Bladder Cancer: FDA-Approved Tests and Emerging Tools for Diagnosis and Surveillance

**DOI:** 10.3390/cancers17213425

**Published:** 2025-10-25

**Authors:** Zhenyun Yang, Fengyu Song, Jin Zhong

**Affiliations:** 1Department of Internal Medicine, School of Medicine, University of California, Riverside, CA 92521, USA; fengyu.song@medsch.ucr.edu; 2Pathology and Laboratory Medicine Service, VA Greater Los Angeles, Los Angeles, CA 90073, USA

**Keywords:** bladder cancer, urinary biomarkers, urine cytology, next-generation sequencing, artificial intelligence

## Abstract

**Simple Summary:**

Bladder cancer is a common cancer with a high risk of recurrence, making early detection and long-term monitoring essential. While cystoscopy remains the diagnostic gold standard, its invasiveness and cost highlight the need for noninvasive urinary biomarkers. Early FDA-approved assays, including Nuclear Matrix Protein 22 (NMP22), Bladder Tumor Antigen Stat and TRAK (BTA Stat/TRAK), Immunocytochemical Assay (ImmunoCyt/uCyt+), and the UroVysion fluorescence in situ hybridization (FISH) test, improved sensitivity but lacked sufficient specificity. Subsequent molecular assays in the 2000s–2010s, including genetic, epigenetic, and RNA-based panels, offered greater diagnostic accuracy and risk stratification. Recent innovations in next-generation sequencing (NGS), exosome analysis, and artificial intelligence (AI)-driven platforms have further advanced noninvasive detection. Despite limitations in sensitivity, specificity, cost, and clinical adoption, urinary biomarkers are increasingly valuable adjuncts to cystoscopy. Emerging multi-omics and computational approaches hold promise for more precise, patient-friendly bladder cancer management.

**Abstract:**

Bladder cancer is a prevalent malignancy with high morbidity and mortality, particularly when diagnosed at an advanced stage. Early detection is critical, as it significantly improves prognosis and the patient’s outcomes. Bladder cancer also has a high recurrence rate, necessitating long-term surveillance. While cystoscopy remains the gold standard for diagnosis and monitoring, it is invasive and costly. Urine cytology, though widely used, has high specificity for detecting high-grade urothelial carcinoma but suffers from low sensitivity and limited effectiveness as a stand-alone diagnostic tool. Urinary biomarkers offer a promising, noninvasive alternative for early detection and disease surveillance. This review examines FDA-approved urinary biomarker tests, including NMP 22, UroVysion, and BTA, highlighting their clinical utility and limitations. Additionally, we explore emerging biomarkers such as DNA methylation assays, genomic alterations, and proteomic signatures as well as advanced technologies like next-generation sequencing and machine learning-based platforms. These innovations have the potential to enhance diagnostic accuracy, risk stratification, and recurrent monitoring, ultimately improving early detection and long-term disease management. By evaluating both established and emerging urinary biomarkers, this review aims to provide clinicians and researchers with insights into evolving tools for bladder cancer diagnosis and surveillance.

## 1. Introduction

Bladder cancer is one of the most common malignancies of the urinary tract, ranking among the top ten cancers worldwide in terms of incidence [1]. It is characterized by a high rate of recurrence and significant risk of progression, making lifelong surveillance necessary for affected patients [2]. The gold standard for diagnosis and follow-up of bladder cancer has traditionally been cystoscopy, an endoscopic technique that allows for direct visualization of the bladder mucosa [3]. Although cystoscopy is highly specific, it is invasive, costly, and often uncomfortable for patients, emphasizing the need for noninvasive diagnostic methods [4].

The earliest attempt to meet this need was urine cytology, introduced in 1945, which became the first widely used urinary biomarker for bladder cancer [5]. By examining exfoliated urothelial cells microscopically, cytology provided a noninvasive method of tumor detection. Although it remains a pivotal tool in clinical practice due to its high specificity for high-grade tumors, its limited sensitivity for low-grade disease pointed to the demand for more advanced biomarkers [6].

Beginning in the early 1990s, a new wave of urinary biomarker assays was developed. These included protein-based tests such as BTA Stat/TRAK and NMP22, which received FDA approval and introduced antigen detection into clinical practice [7,8]. Following that, ImmunoCyt/uCyt+ and UroVysion FISH were developed by targeting cell-surface antigens and chromosomal abnormalities, respectively [9,10].

The 2000s brought about an advancement of molecular biomarkers focusing on telomerase activity, microsatellite instability, and DNA methylation patterns [11,12,13]. In the 2010s, multiparametric molecular platforms such as Cxbladder and AssureMDx integrated gene expression and mutational analyses into clinical testing [14,15].

Most recently, the field has moved toward next-generation approaches, including liquid biopsy panels detecting recurrent mutations (e.g., FGFR3, TERT promoter, TP53), exosomal cargo profiling, and multi-omics platforms incorporating transcriptomics, epigenomics, proteomics, and metabolomics that are enhanced by artificial intelligence [16,17,18,19]. These advances represent a remarkable evolution from simple cytological observation to complex molecular interrogation, reflecting both the challenges and the progress in developing reliable urinary biomarkers for bladder cancer.

In this review, we provide a chronological overview of bladder cancer biomarkers, from cytology to modern molecular platforms, with particular emphasis on emerging technologies such as next-generation sequencing and machine learning.

## 2. Methodology

This work is a narrative review summarizing recent advances in urinary biomarkers for bladder cancer. Relevant studies were identified through literature search in PubMed using terms such as bladder cancer, urinary biomarkers, urine cytology, NGS, and artificial intelligence. Articles were selected for their relevance to biomarker development, diagnostic performance, and clinical applicability.

## 3. 1945: Urine Cytology

Urine cytology, first described in 1945 by Papanicolaou and Marshall [5], represents the earliest and most widely used noninvasive urinary biomarker for bladder cancer. The method rapidly gained clinical traction and was further established in the mid-20th century through studies that characterized the cytologic features of bladder tumors [20]. By examining exfoliated urothelial cells microscopically, cytology provided a noninvasive means of tumor detection, with malignant cells recognized by abnormal nuclear morphology, high nuclear-to-cytoplasmic ratios, and irregular chromatin patterns [21,22].

The major strength of urine cytology lies in its high specificity, particularly for high-grade urothelial carcinoma and carcinoma in situ, where diagnostic accuracy approaches 90–100% in some series [6,23]. However, its sensitivity is markedly limited for low-grade tumors, since these cells often resemble normal urothelium and shed less frequently into urine [24]. Additional drawbacks include inter-observer variability, dependence on pathologist expertise, and reduced performance in cases with inflammation or atypical findings [25].

Despite these limitations, urine cytology remains an essential adjunct to cystoscopy in bladder cancer diagnosis and follow-up, and it continues to serve as the benchmark against which newer urinary biomarkers are compared [6,26].

## 4. Urinary Biomarkers Developed in the 1990s: The Era of FDA-Approved Assays

The 1990s marked a pivotal era in urinary biomarker development, leading to multiple FDA-approved assays for bladder cancer detection. In the early 1990s, several urinary biomarker assays were developed to address the limitations of cytology. Among them, BTA Stat and BTA TRAK were first developed, which detect human complement factor H-related proteins released by bladder tumor cells. These tests provided a rapid, point-of-care approach, and although they improved sensitivity compared to cytology, their specificity was reduced due to false positives from hematuria and inflammation [27].

Another milestone was the introduction of NMP22, a nuclear matrix protein released during urothelial cell apoptosis and necrosis [28]. Both an ELISA-based format (NMP22 Test Kit) and a point-of-care version (NMP22 BladderChek) were FDA-approved. These assays demonstrated higher sensitivity than cytology, particularly for recurrent disease [8]. However, like BTA, specificity was compromised in patients with benign urological conditions such as urinary tract infections and stones [29].

To enhance detection accuracy, ImmunoCyt/uCyt+ was developed in the late 1990s. This assay uses fluorescently labeled monoclonal antibodies to detect bladder cancer-associated antigens such as CEA and mucin-like glycoproteins in exfoliated urothelial cells. It demonstrated significantly higher sensitivity than cytology, especially for low-grade tumors, though its requirement for specialized fluorescence microscopy limited widespread adoption [30].

UroVysion fluorescence in situ hybridization (FISH) was introduced around 2000, targeting chromosomal abnormalities commonly seen in bladder cancer (e.g., aneuploidy of chromosomes 3, 7, 17, and deletion of 9p21). FDA-approved for use in both diagnosis and surveillance, UroVysion offered strong sensitivity for detecting recurrent high-grade disease, though false positives and high costs restricted routine use [31].

Another urinary biomarker assay developed in 1998 is the Cytokeratin 20 (CK20) assay, which is not FDA-approved. This assay detects the expression of CK20, a protein commonly found in urothelial cells and overexpressed in bladder cancer [32]. Detection is typically performed through immunocytochemistry or reverse transcription PCR on urine samples. Elevated CK20 expression serves as an indicator of malignant urothelial cells, offering a noninvasive tool for diagnosis and surveillance. However, its sensitivity can vary, particularly in low-grade tumors, and false positives may occur in inflammatory conditions.

Table 1 summarizes the first clinically adopted urinary biomarker assays of the 1990s, including primarily protein-based tests (BTA, NMP22) and immunocytochemical assays (ImmunoCyt), excluding the CK20 assay as it is not FDA-approved. Collectively, these FDA-approved tests represented a paradigm shift in bladder cancer diagnostics, introducing antigen and chromosomal marker-based detection into clinical practice. While none replaced cystoscopy, they laid the foundation for molecular assays that would emerge in the 2000s and beyond.

## 5. 2000s: Molecular Biomarkers

The 2000s marked a transition from antigen-based assays toward molecular biomarker approaches that leveraged advances in genomics and molecular biology. Among the earliest investigated was telomerase activity, an enzyme upregulated in most cancers to maintain replicative potential. Several studies demonstrated that telomerase activity in urine and bladder washings had high sensitivity for urothelial carcinoma, particularly in detecting early-stage and recurrent tumors [33,34].

Microsatellite instability (MSI) and loss of heterozygosity (LOH) were important categories. These genomic alterations, especially at chromosome 9p21 and other loci, were detectable in exfoliated urinary cells and showed promise as markers for recurrence monitoring [35,36]. Although highly informative in research, technical complexity and cost limit their translation to routine practice.

DNA methylation-based assays emerged as particularly promising in this decade. Aberrant promoter methylation of tumor suppressor genes such as RASSF1A, APC, and CDKN2A was shown to be frequent in bladder cancer and detectable in voided urine [37,38]. Early clinical studies demonstrated high sensitivity and specificity, positioning methylation as a leading candidate for noninvasive surveillance [39,40,41].

These molecular biomarkers (summarized in Table 2) represented a conceptual leap, shifting bladder cancer diagnostics toward genomic and epigenetic profiling. While not yet widely adopted clinically at the time, they built the foundation for the multi-gene methylation panels and integrated molecular assays that became prominent in the 2010s.

## 6. 2010s: Multiparametric Molecular Panels (RNA, DNA, and Methylation Assays)

The 2010s saw the rise of multiparametric molecular platforms for noninvasive bladder cancer detection, leveraging RNA expression, DNA mutations, and epigenetic modifications (summarized in Table 3). These assays aimed to improve sensitivity and specificity over traditional protein- or cytology-based tests while capturing molecular heterogeneity of tumors [43,44,45,46,47,48,49].

Cxbladder, a urine-based mRNA panel, integrates five gene targets (IGFBP5, HOXA13, MDK, CDK1, CXCR2) implicated in tumorigenesis, cell cycle regulation, and inflammation. Clinical studies demonstrated that Cxbladder could stratify patients into high- and low-risk groups, improving evaluation of hematuria and monitoring for recurrence [43,44,45]. Its sensitivity for low-grade tumors is higher than urine cytology, although specificity can be reduced in patients with benign urinary conditions [44,46].

AssureMDx analyzes DNA from exfoliated urothelial cells in urine to identify mutations (FGFR3, TERT, HRAS) and methylation changes (OTX1, ONECUT2, TWIST1), allowing for highly accurate detection of bladder cancer. This multi-layered approach leverages both genetic and epigenetic alterations, improving detection of primary and recurrent disease [47,48,49]. While not FDA-approved, it has been increasingly adopted as a laboratory-developed test in specialized centers under CLIA-certified protocols [48].

DNA methylation panels gained prominence in this decade, as promoter hypermethylation is a common epigenetic event in bladder cancer [50,51,52,53]. Markers such as TWIST1, NID2, OTX1, and SIM2 were repeatedly validated to distinguish malignant from benign urinary samples, providing a noninvasive alternative to cystoscopy for surveillance and early detection [51,52,53]. Streamlined assays, such as the EarlyTect BCD PENK methylation test, exemplify how DNA methylation diagnostics have evolved from initial research into clinically viable tests [52,54].

Additionally, combined multi-gene panels and RNA expression signatures have been explored for risk stratification and predicting recurrence or progression [3,20,55,56]. These platforms represent a shift from single analyte testing toward integrative molecular diagnostics, paving the way for multi-omics strategies that further enhance diagnostic power in the following decade [3,20,56].

**Table 3 cancers-17-03425-t003:** Urinary Biomarker Assays in the 2010s—Multi-Marker Panels and Methylation Signatures.

Assay	Year Introduced	Principle	FDA/CE Status	Key Notes
Cxbladder	2013	Measures expression of 5 mRNA genes (MDK, HOXA13, CDC2, IGFBP5, CXCR2)	LDT (NZ, US)	High sensitivity; used for both diagnosis and surveillance [44,45]
Bladder EpiCheck	2016	DNA methylation panel of 15 genomic loci	CE-marked (EU)	Robust negative predictive value for recurrence [54]
Xpert Bladder Cancer Monitor	2017	qRT-PCR detecting 5 bladder cancer mRNAs	CE-marked (EU)	Cartridge-based, automated, rapid workflow [56]
AssureMDx	2017	Detects FGFR3, HRAS, TERT mutations plus methylation markers	LDT	High NPV in hematuria patients; used to triage cystoscopy [47]
UroMark	2010s (UK research)	High-throughput methylation panel (>150 loci)	Research only	Developed as a large-scale epigenetic screening assay [51]
ADx Bladder Test	2010s (France)	Detects specific mRNA biomarkers in urine	CE-marked	Used in Europe for surveillance and follow-up [56]

## 7. 2020s: Advancements in Bladder Cancer Biomarkers (NGS, Exosome-Based, and AI Integration)

The 2020s have been accompanied by significant advancements in bladder cancer diagnostics, propelled by innovations in next-generation sequencing (NGS), exosome-based analyses, and artificial intelligence (AI). These technologies aim to enhance diagnostic accuracy, enable early detection, and support personalized treatment strategies [42,57,58,59,60,61,62,63,64,65,66,67,68].

NGS has transformed noninvasive bladder cancer diagnostics by enabling comprehensive analysis of multiple genetic alterations in urine-derived DNA. Urinary NGS panels can simultaneously detect mutations in genes commonly altered in bladder tumors, including TERT promoter, FGFR3, TP53, PIK3CA, and ERBB2 [42,57]. TERT promoter mutations are present in approximately 55% of bladder cancers across all stages, while FGFR3 mutations are found in 30–50% of cases, often in low-grade, non-muscle-invasive tumors [42,57]. PIK3CA mutations occur in roughly 35% of cases, TP53 in 24%, and ERBB2 amplifications in a smaller subset typically associated with aggressive disease [42,57,58]. By capturing these alterations in urine, NGS offers a noninvasive method for early diagnosis, tumor monitoring, and longitudinal tracking of clonal evolution.

NGS-based urinary assays have demonstrated high sensitivity and specificity, with some studies reporting diagnostic accuracies above 90% [42,58]. Multi-gene panels outperform single-gene approaches by improving the detection of low-grade tumors and capturing tumor heterogeneity [42,58].

Urinary exosomes, nanoscale extracellular vesicles secreted by tumor cells, have emerged as a rich source of bladder cancer biomarkers. Exosomes carry proteins, DNA fragments, and RNA species, including microRNAs (miRNAs), that reflect the molecular characteristics of the tumor. Studies have identified exosomal miRNAs such as miR-21 and members of the miR-200 family, as well as proteins like survivin, with high diagnostic accuracy for bladder cancer [57,59]. Exosomes are particularly attractive because their contents are stable in urine, allowing noninvasive sampling and repeated monitoring [57,59]. Preliminary studies suggest that exosomal assays may detect early-stage disease with higher sensitivity than conventional protein- or cytology-based tests and may also provide insights into tumor aggressiveness and recurrence risk [57,59].

The integration of AI into bladder cancer diagnostics has seen significant advancements in the 2020s, particularly in the analysis of urinary biomarkers and cystoscopic imaging [58,59,60,61,62,63,64,65]. These developments aim to enhance diagnostic accuracy, predict disease progression, and personalize treatment strategies [58,59,60,61,62,63,64,65].

AI algorithms have been applied to analyze complex datasets derived from urinary biomarkers, including miRNA profiles and clinical data. For instance, a study by Moisoiu et al. utilized machine learning models—Random Forest, Support Vector Machine (SVM), and XGBoost—to integrate miRNA expression levels with demographic and clinical data, demonstrating improved diagnostic accuracy for bladder cancer [58,61].

Furthermore, AI has been employed to process droplet patterns in blood and urine samples. A study by Demir et al. developed a deep learning model using ResNet-18 architecture, achieving diagnostic accuracies of 97.3% for blood samples and 95.3% for urine samples [62].

AI has also been integrated into cystoscopic imaging to improve the detection and classification of bladder tumors [64]. A study by Amaouche et al. proposed a hybrid convolutional neural network (CNN)-transformer model for bladder cancer detection and segmentation, demonstrating high diagnostic accuracy [64].

Additionally, a multi-task deep learning framework developed by Yu et al. combined EfficientNet-B0 for classification, ResNet34-UNet++ for segmentation, and ConvNeXt-Tiny for molecular subtyping, achieving an accuracy of 93.28%, F1-score of 82.05%, and AUC of 96.41% [63].

AI models have been utilized to predict the recurrence and progression of non-muscle-invasive bladder cancer (NMIBC). A study by Abbas et al. employed the Tsetlin Machine, an interpretable AI model, to analyze clinical features and predict NMIBC recurrence, achieving an F1-score of 0.80. The F1-score combines precision and recall reflecting the balance between correct predictions and missed cases; a value of 0.8 indicates strong overall model performance and reliability in predicting recurrence [65].

In summary, the 2020s have witnessed a rapid evolution in bladder cancer diagnostics, driven by the integration of next-generation sequencing, exosome-based analyses, and artificial intelligence (summarized in Table 4) [42,57,58,59,60,61,62,63,64,65,66,67,68]. These approaches enable highly sensitive and noninvasive detection of genetic mutations, epigenetic changes, and tumor-derived extracellular vesicles, capturing the molecular heterogeneity of bladder cancer more comprehensively than ever before [42,57,58,59,60,61,62,63,64,65,66,67,68]. AI-powered models further enhance diagnostic performance by integrating complex datasets from urinary biomarkers, clinical features, and imaging, allowing for accurate tumor classification, risk prediction, and monitoring of disease recurrence or progression [42,57,58,59,60,61,62,63,64,65,66,67,68]. Collectively, these innovations provide a multi-dimensional, personalized framework that holds promise for improving early detection, guiding treatment decisions, and enabling precision surveillance, ultimately transforming the management and long-term outcomes of bladder cancer patients [42,57,58,59,60,61,62,63,64,65,66,67,68].

## 8. Discussion

### 8.1. Current Status of Bladder Cancer Biomarkers

Urinary biomarkers have become an increasingly valuable tool in the noninvasive detection and surveillance of bladder cancer. They offer several advantages, including the potential to identify tumors that may be missed by standard urine cytology, reduce reliance on invasive cystoscopy, and enable repeated, convenient monitoring through simple urine collection. Many biomarkers detect molecular or cellular alterations associated with malignancy, including proteins shed into the urine, genetic mutations, epigenetic changes, and extracellular vesicles such as exosomes. These assays can provide insights not only into the presence of disease but, in some cases, also into tumor biology and risk of recurrence [57].

Urinary biomarkers also have important limitations. Sensitivity and specificity can vary depending on tumor grade, stage, and the presence of benign urological conditions such as infection, inflammation, or hematuria, which may cause false-positive or false-negative results. Some assays require specialized laboratory equipment or technical expertise, limiting accessibility in routine clinical practice. Variability in assay performance across different laboratories and platforms can affect reliability, and many tests still require further clinical validation to establish consistent accuracy and predictive value. Cost and reimbursement considerations may also influence their practical implementation, especially for complex multi-analyte or omics-based platforms. Despite these challenges, urinary biomarkers represent a promising adjunct to conventional diagnostic methods, offering the potential to improve early detection, risk stratification, and longitudinal monitoring of bladder cancer patients [60].

### 8.2. FDA-Approved Versus Non-FDA-Approved Biomarkers in Clinical Adoption

FDA clearance has historically represented an important milestone in the development of urinary biomarkers, but regulatory approval alone has not guaranteed routine clinical adoption. Early FDA-approved assays such as BTA, NMP22, ImmunoCyt/uCyt+, and UroVysion established analytical validity and clinical feasibility, yet their uptake has been modest because of moderate diagnostic accuracy and limited insurance coverage [8,28,29,30,31,32].

In contrast, several non-FDA-approved but CLIA-validated assays, particularly those based on gene mutations, DNA methylation, and RNA expression, have achieved increasing clinical use [44,45,48,54,56]. These laboratory-developed tests, including multi-gene and next-generation sequencing-based panels, often outperform earlier FDA-approved assays in sensitivity and specificity, especially for detecting low-grade or recurrent disease.

Ultimately, clinical adoption has been more strongly influenced by reimbursement policies, evidence of clinical utility, and cost-effectiveness than by regulatory designation alone. As such, the biomarker field now includes both FDA-cleared and non-cleared assays used in practice, reflecting a pragmatic balance between regulatory validation and clinical performance.

It is important to note, however, that obtaining FDA approval for newer molecular biomarkers is inherently time-consuming and requires extensive analytical validation, multicenter clinical trials, and demonstration of reproducible diagnostic accuracy across diverse patient populations. Many promising assays—particularly those using next-generation sequencing, exosomal profiling, or AI-based algorithms—are still in the investigational or laboratory-developed test phase [42,57,58,59]. As evidence accumulates from real-world studies and large-scale clinical validations, these emerging biomarkers may gradually progress toward regulatory clearance and broader clinical integration.

### 8.3. Traditional Versus Emerging Biomarkers in Bladder Cancer Diagnosis

The evolution of urinary biomarkers for bladder cancer exemplifies the broader transformation from single-analyte assays to integrative, data-driven diagnostic platforms. Traditional biomarkers such as nuclear matrix protein 22 (NMP22), bladder tumor antigen (BTA) Stat/TRAK, ImmunoCyt/uCyt+, and UroVysion fluorescence in situ hybridization (FISH) were designed to detect specific tumor-associated proteins or chromosomal abnormalities shed into urine [8,9,10,11]. These assays provided early proof of concept for noninvasive diagnosis and surveillance and remain important adjuncts to cystoscopy. However, their clinical performance has been limited by moderate sensitivity, false positives in inflammatory or benign urological conditions, and an inability to fully capture tumor heterogeneity [28,29,30,31,32].

Emerging urinary biomarkers, in contrast, utilize genomic, transcriptomic, and epigenomic information to provide a more comprehensive molecular characterization of bladder cancer. Modern platforms detect driver mutations such as TERT promoter, FGFR3, TP53, and PIK3CA, and assess DNA methylation changes in tumor suppressor genes including RASSF1A, TWIST1, and OTX1 [33,34,35,36,37,38,47,48,49,51,52,53,54]. In addition, RNA- and microRNA-based signatures, as well as exosome-derived analytes, reflect transcriptional and post-transcriptional regulation associated with tumor initiation, progression, and recurrence [19,20,42,55,56,57,58,59]. These multi-analyte approaches markedly improve sensitivity and specificity, particularly for low-grade and early-stage disease, while offering the potential for longitudinal disease monitoring.

From an analytical standpoint, traditional assays typically generate qualitative or semi-quantitative outputs, whereas emerging technologies incorporate machine learning and bioinformatics to interpret high-dimensional datasets. Such computational frameworks enable the identification of subtle molecular patterns and predictive signatures, facilitating tumor classification, recurrence risk prediction, and treatment response assessment [58,59,60,61,62,63,64,65]. This integration of molecular data and artificial intelligence represents a paradigm shift, positioning urinary biomarkers as tools for precision oncology rather than mere detection assays.

Despite these advances, several challenges must be addressed before widespread clinical adoption. Emerging biomarkers require large-scale validation, standardized assay protocols, and robust evidence of clinical utility and cost-effectiveness. Achieving regulatory approval, such as through the U.S. Food and Drug Administration, also demands reproducibility across diverse populations and laboratories. As additional prospective and multicenter studies continue to demonstrate consistent diagnostic accuracy, recent assays—particularly those targeting TERT promoter mutations, DNA methylation panels, and exosomal RNA—are likely to progress toward regulatory clearance and incorporation into clinical practice [42,57,58,59].

Collectively, the field is shifting from protein- and cytology-based detection toward multi-omics integration and computational analytics, providing a more complete understanding of tumor biology and enabling earlier, more accurate, and individualized diagnosis of bladder cancer.

### 8.4. Continuity of Biomarker Development

Although we have described urinary biomarkers in a chronological framework, from cytology in the 1940s to multi-omics and AI integration in the 2020s, it is important to recognize that biomarker development is a continuous and evolving process rather than a series of discrete eras. Innovations from the 2000s and 2010s have not become obsolete; rather, they provide the foundation for current and emerging technologies. For example, DNA methylation assays, which first gained attention in the 2010s, continue to evolve with improved sensitivity, streamlined workflows, and broader clinical applicability. The EarlyTect BCD PENK methylation test exemplifies this progression, integrating lessons from earlier methylation studies into a simplified, highly sensitive urine-based assay that achieves excellent performance across diverse populations [69]. This demonstrates how foundational research in previous decades seamlessly extends into modern diagnostics, highlighting a continuum of innovation with ongoing potential to enhance early detection, monitoring, and personalized management of bladder cancer.

## 9. Conclusions

Urinary biomarkers have become an increasingly important component of bladder cancer diagnosis and surveillance, offering a noninvasive alternative to cystoscopy and complementing traditional urine cytology. They provide the ability to detect tumors earlier, monitor disease recurrence, and gain molecular insights into tumor biology, which can guide personalized patient management [57,60]. Traditional biomarkers, such as NMP22, BTA, ImmunoCyt, and UroVysion FISH, established the foundation for non-invasive testing by targeting single proteins or chromosomal abnormalities [58]. Emerging platforms, including multi-gene panels, DNA methylation assays, exosome profiling, and integrated multi-omics approaches enhanced by artificial intelligence, now allow for more comprehensive molecular characterization, higher sensitivity, and improved detection across tumor grades.

Despite these advantages, urinary biomarkers still face important limitations. Sensitivity and specificity can vary depending on tumor stage, grade, and the presence of benign urological conditions, while technical complexity, cost, and variability across laboratories can limit widespread clinical implementation [57,58]. Many emerging assays also require further validation before routine adoption, and reimbursement policies continue to influence their practical use [59,60,61,62,63,64,65,66,67,68,69].

Nonetheless, the future of urinary biomarker research is highly promising. Advances in multi-omics integration, liquid biopsy technologies, and AI-driven predictive modeling have the potential to transform bladder cancer care by enabling precision diagnostics, real-time risk stratification, and personalized surveillance strategies. As these technologies mature, they are expected to reduce reliance on invasive procedures, improve early detection, and ultimately enhance patient outcomes, representing a significant step forward in the management of bladder cancer [57,60,64,65,66,67,68].

## Figures and Tables

**Table 1 cancers-17-03425-t001:** Urinary Biomarker Assays in the 1990s—First Generation Protein-Based and Cell-Based Tests.

Assay	Year Introduced/Approved	Principle	FDA Status	Key Notes
BTA Stat	Early 1990s	Detects complement factor H-related proteins in urine	FDA-approved (point-of-care—POC)	Rapid immunoassay; higher sensitivity than cytology, but false positives with hematuria/inflammation [27]
BTA TRAK	Early 1990s	ELISA detecting complement factor H-related proteins	FDA-approved (lab-based)	Quantitative version of BTA Stat; improved sensitivity but limited specificity [27]
NMP22 (ELISA)	1996	Detects nuclear mitotic apparatus protein released during cell death	FDA-approved (lab-based)	Useful for recurrence monitoring; false positives with benign urological conditions [29]
NMP22 BladderChek	Late 1990s	Point-of-care immunoassay detecting NMP22	FDA-approved (POC)	Quick test; widely studied in follow-up surveillance [8]
ImmunoCyt/uCyt+	Late 1990s	Fluorescent monoclonal antibodies against bladder tumor-associated antigens (CEA, mucin-like glycoproteins)	FDA-approved (adjunct to cytology)	Improved sensitivity for low-grade tumors; require fluorescence microscopy [30]
UroVysion FISH	2000	Multicolor FISH detecting aneuploidy of chromosomes 3, 7, 17, and 9p21 deletion [10]	FDA-approved	Gold standard molecular cytogenetics assay; strong for high-grade and equivocal cytology [31]

**Table 2 cancers-17-03425-t002:** Urinary Biomarker Assays in the 2000s—Genetic and Chromosomal Marker Era.

Assay	Year Introduced	Principle	FDA Status	Key Notes
UroVysion FISH	2001	Multicolor FISH detecting aneuploidy of chromosomes 3, 7, 17, and 9p21 deletion	FDA-approved	Gold standard molecular cytogenetics assay; strong for high-grade and equivocal cytology [32]
Microsatellite Analysis	Early 2000s (research)	Detects loss of heterozygosity (LOH) in urine DNA	Research only	Pioneered DNA-based urinary diagnostics; labor intensive [35]
hTERT Promoter Mutations	Early 2000s (research)	PCR-based detection of telomerase reverse transcriptase mutations	Research only	One of the most frequent alterations in bladder cancer [11]
FGFR3 Mutation Assays	2000s (research)	Detects FGFR3 hotspot mutations in urine	Research only	Associated with low-grade, non-muscle-invasive bladder cancers [42]
Survivin Assay	2000s (research)	Detects survivin protein/mRNA in urine	Research only	Marker of apoptosis inhibition; promising but variable performance [30]
CYFRA 21-1	2000s	Detects soluble cytokeratin 19 fragments in urine	Research only	Evaluated in several studies; modest diagnostic accuracy [30]

**Table 4 cancers-17-03425-t004:** Urinary Biomarker Assays in the 2020s—Advanced Genomics, Exosomes, and AI Integration.

Assay	Year Introduced	Principle	FDA/CE Status	Key Notes
UroSEEK	2020s	NGS detecting mutations (TERT, FGFR3, TP53, others) and aneuploidy	Clinical trials	High sensitivity; integrates multiple molecular alterations [56]
ExoDx (ExosomeDx) Bladder Test	2020s	Exosomal RNA and protein profiling	Research/LDT	Leverages extracellular vesicles as tumor carriers [59]
OncoUrine	2020s	Targeted NGS + urine cfDNA analysis	CE-marked (China/EU)	Provides recurrence monitoring, expanding global use [42]
AI-enhanced Digital Cytology	2020s	Deep learning applied to cytology slides + molecular inputs	Research/early adoption	Improves accuracy of urine cytology; integrated workflows emerging [64]
EarlyTect BCD	2024	PENK gene methylation assay in urine DNA	CE-marked (Korea)	Single-gene streamlined test; validated in hematuria patients [69]
miRNA Panels (e.g., miR-126, miR-146a, miR-200 family)	2020s (research)	Urinary exosomal or free miRNA profiling	Research only	Promising biomarkers, under clinical validation [61]

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
