# Peer review of "Urinary Biomarkers in Bladder Cancer: FDA-Approved Tests and Emerging Tools for Diagnosis and Surveillance"

_cancers, 2025, doi:10.3390/cancers17213425_

Round 1
Reviewer 1 Report
Comments and Suggestions for Authors
The review entitled "Urinary Biomarkers in Bladder Cancer: FDA-Approved Tests and Emerging Tools for Diagnosis and Surveillance" describes the tools used for bladder cancer diagnosis and surveillance. Bladder cancer is the ninth most common cancer worldwide and early monitoring is essential for proper diagnosis. In this review, the authors have tried to summarize the various biomarkers used for its detection. However, the overall structure and organization of the review require significant improvement.
There are many grammatic errors and typos in the article.
The section headings "FDA-approved Versus Non-FDA-approved Biomarkers in Clinical Adoption" and "Traditional Versus Emerging Biomarkers in Bladder Cancer Diagnosis" are repetitions of what has been mentioned in other sections even though the authors tried to explain with the clinical guidelines.
In addition, the full form of the tests were not given in the beginning.
The term "AssureMDx" is not explained.
Line 28: change “low sensitivity, limited its effectiveness” to “low sensitivity and limited effectiveness”
Line 30: typo “biomakers tests” to “biomarker tests”
Full forms of the terms BTA Stat/TRAK and NMP22 missing
Comments on the Quality of English Language
There are a number of instances of text repetition.
Author Response
Authors agree with this comment and have made the change accordingly. Thank you very much for taking time to help us to modify the manuscript!
Comments and Suggestions for Authors
The review entitled "Urinary Biomarkers in Bladder Cancer: FDA-Approved Tests and Emerging Tools for Diagnosis and Surveillance" describes the tools used for bladder cancer diagnosis and surveillance. Bladder cancer is the ninth most common cancer worldwide and early monitoring is essential for proper diagnosis. In this review, the authors have tried to summarize the various biomarkers used for its detection. However, the overall structure and organization of the review require significant improvement.
The manuscript has been modified significantly - we added a methodology section, rewrote most of the discussion part, and made lots of other changes.
There are many grammatic errors and typos in the article.
Thanks for pointing out. We have double checked and make numerous changes to correct the typos and grammatic errors – highlighted in the red in the manuscript.
The section headings "FDA-approved Versus Non-FDA-approved Biomarkers in Clinical Adoption" and "Traditional Versus Emerging Biomarkers in Bladder Cancer Diagnosis" are repetitions of what has been mentioned in other sections even though the authors tried to explain with the clinical guidelines.
We have rewritten these two parts in the discussion to minimize the repetitions.
In addition, the full forms of the tests were
not given in the beginning.
The full forms of the tests are given now – in red.
The term "AssureMDx" is not explained.
A paragraph was added to explain AssureMDx in the 2010s section.
Line 28: change “low sensitivity, limited its effectiveness” to “low sensitivity and limited effectiveness”
Change has been made.
Line 30: typo “biomakers tests” to “biomarker tests”
Change has been made.
Full forms of the terms BTA Stat/TRAK and NMP22 missing
The full forms of the tests are given now – in red.
Comments on the Quality of English Language
There are a number of instances of text repetition.
We tried to remove all these repetitions.
Reviewer 2 Report
Comments and Suggestions for Authors
This is a well-written and easy-to-read narrative review on urinary bladder biomarkers. It presents an interesting and concise summary of the current state of the art. However, there are several issues that should be addressed:
-
A brief description of the methodology should be included. The authors should clarify that this is a narrative (not a systematic) review and provide some details on how the selected papers were identified and chosen for inclusion.
-
The Cytokeratin 20 (CK20) assay, which is not FDA-approved, should be removed from the first group of markers and also from Table 1, as the table is intended to include only FDA-approved markers.
-
Additionally, CK20 is not clearly referenced in the text, and there are no specific comments about it outside of its inclusion in the table. This inconsistency should be addressed.
-
The information presented in the tables should be properly referenced. Each biomarker mentioned should include an appropriate citation.
Author Response
Authors agree with this comment and have made the change accordingly. Thank you very much for taking time to help us to modify the manuscript!
This is a well-written and easy-to-read narrative review on urinary bladder biomarkers. It presents an interesting and concise summary of the current state of the art. However, there are several issues that should be addressed:
- A brief description of the methodology should be included. The authors should clarify that this is a narrative (not a systematic) review and provide some details on how the selected papers were identified and chosen for inclusion.
A methodology section was added.
- The Cytokeratin 20 (CK20) assay, which is not FDA-approved, should be removed from the first group of markers and also from Table 1, as the table is intended to include only FDA-approved markers.
The CK20 was removed from table and specifically pointed out that it is not FDA-approved.
- Additionally, CK20 is not clearly referenced in the text, and there are no specific comments about it outside of its inclusion in the table. This inconsistency should be addressed.
A paragraph about CK20 was added in the 1990s section. Since it is not FDA-approved, it was removed from table 1.
- The information presented in the tables should be properly referenced. Each biomarker mentioned should include an appropriate citation.
Citations were included in the tables.
Round 2
Reviewer 1 Report
Comments and Suggestions for Authors
The authors have addressed all comments and suggestions in the revised manuscript.
Reviewer 2 Report
Comments and Suggestions for Authors
All the comments have been addressed